# Generalizing Consistency Policy to Visual RL with Prioritized Proximal Experience Regularization

**Haoran Li**
Institute of Automation,
Chinese Academy of Sciences
University of Chinese Academy of Sciences
lihaoran2015@ia.ac.cn

**Zhennan Jiang**
Institute of Automation,
Chinese Academy of Sciences
University of Chinese Academy of Sciences
jiangzhennan2024@ia.ac.cn

**Yuhui Chen**
Institute of Automation,
Chinese Academy of Sciences
University of Chinese Academy of Sciences
chenyuhui2022@ia.ac.cn

**Dongbin Zhao**[*]
Institute of Automation,
Chinese Academy of Sciences
University of Chinese Academy of Sciences
dongbin.zhao@ia.ac.cn

## Abstract

With high-dimensional state spaces, visual reinforcement learning (RL) faces significant challenges in exploitation and exploration, resulting in low sample efficiency and training stability. As a time-efficient diffusion model, although consistency models have been validated in online state-based RL, it is still an open question whether it can be extended to visual RL. In this paper, we investigate the impact of non-stationary distribution and the actor-critic framework on consistency policy in online RL, and find that consistency policy was unstable during the training, especially in visual RL with the high-dimensional state space. To this end, we suggest sample-based entropy regularization to stabilize the policy training, and propose a consistency policy with prioritized proximal experience regularization (CP3ER) to improve sample efficiency. CP3ER achieves new state-of-the-art (SOTA) performance in 21 tasks across DeepMind control suite and Meta-world. To the best of our knowledge, CP3ER is the first method to apply diffusion/consistency models to visual RL and demonstrates the potential of consistency models in visual RL. Our project page is hosted at https://jzndd.github.io/CP3ER-Page/.

## 1 Introduction

RL has achieved remarkable results in many fields, such as video games [1], Go [2], Chess [3, 4] and robotics [5–8]. Since it is hard to parameterize the complex policy distribution over high-dimensional state and continuous action spaces, the performance and stability of visual RL are still unsatisfactory. As the most common policy distribution, Gaussian distribution is easy to sample, but its unimodal nature limits the expressiveness to represent complex behaviors [9]. While complex distributions have the rich, expressive power to improve the exploration ability [10], the difficulty of sampling makes it hard to apply to online RL directly. Parameterizing the complex policy distribution to balance ease of sampling and expressiveness is a bottleneck to improving the efficiency of visual RL.

As an emerging generation model, the diffusion model [11] stands out in fields such as image generation [12–14] and video generation [15, 16] with its ability to model complex distributions and ease of sampling characteristics. These properties have also been explored for learning a complex

---

[*]Corresponding author.

38th Conference on Neural Information Processing Systems (NeurIPS 2024).

policy [17]. For example, diffusion models are used to imitat e the diverse expert policies [18, 19] or trajectories [20–23] in datasets. In addition, due to their excellent expressive and data generation abilities, diffusion models are often employed to address policy constraints [24–26] and data scarcity [21, 27, 28] in offline RL. Most of these applications are limited to offline learning due to the demand for pre-collected datasets to train diffusion models.

Applying diffusion models in online RL will face different problems than offline RL. Firstly, unlike pre-collected data in offline RL, the data distribution in online RL is non-stationary [29], and it is currently unclear whether this change will impact training diffusion models. Secondly, since the optimal policy distribution is unknown, samples from this distribution are inaccessible, resulting in the ill-posed traditional score matching problem [30]. In addition, the time-inefficiency of diffusion models [31] will become more prominent with a large number of online interactions, leading to unacceptable time costs for online learning. As an efficient diffusion model, the consistency model [32] directly establishes a mapping from noise to denoised data, which is employed for online RL and achieves time efficiency and better performance [33, 34]. These methods simply replace the Gaussian model in the actor-critic framework with the consistency model and train consistency policy with the Q-loss. Although they achieve competitive performance in state-based RL tasks, this training method is incompatible with traditional score matching for diffusion models. Therefore, the question is whether this training framework is suitable for consistency model-based policy training, especially for visual RL tasks with high-dimensional state spaces.

In this paper, we investigate the impact of non-stationary dataset and the actor-critic framework on consistency policy. By analyzing the dormant ratio [29] of the policy network, we find that the non-stationary of training data is not the main factor affecting the instability of consistency policy, while the Q-loss in the actor-critic framework leads to a sharp increase in the dormant ratio of the policy network, resulting in the loss of complex expression ability, which is particularly significant in visual RL tasks. To address the above issues, we suggest sample-based entropy regularization to stabilize the policy training and propose the prioritized proximal experience regularization, which uses weighted sampling to construct an appropriate proxy policy for policy regularization and achieves sample-efficiency consistency policy. Overall, our contributions are as follows:

- We investigate the impact of non-stationary distribution and actor-critic framework on consistency policy in online RL, and find that the Q-loss of the actor-critic framework can impair the expressive ability of the consistency model, leading to unstable policy training. This phenomenon is particularly significant in visual RL tasks.

- We suggest sample-based entropy regularization and propose a consistency policy with prioritized proximal experience regularization (CP3ER) which significantly enhances the stability of policy training with the Q-loss under the actor-critic framework.

- Our proposed method performs new SOTA in 21 visual control tasks, including DeepMind control suite and Meta-world tasks. To our knowledge, our proposed CP3ER is the first method to apply diffusion/consistency models to visual RL.

## 2 Related Work

### 2.1 Diffusion Model in Reinforcement Learning

Due to its high-quality sample generation ability and training stability, diffusion models [11] have been widely applied in fields such as image generation, video generation, and natural language processing and have also been promoted in RL. Since the diffusion model can represent complex distribution in datasets, it is commonly used in offline RL to model behavior policies [18, 25, 35] or expected policies [34, 36–38] to meet the requirements of diversity policy constraints and achieve a balance between constraint and exploitation. The diffusion model can also model trajectory distribution [20, 39, 40], achieving specified trajectory generation under different guidance. In addition, diffusion models are also employed to generate data to augment limited training data [27, 28].

Although diffusion models have been widely applied in offline learning, using diffusion models in online learning remains a challenging problem. [41] proposes the concept of action gradient, which uses a value function to estimate the gradient of actions and updates the actions in the replay buffer. The diffusion model-based policy is trained based on the updated actions. [42] employs a diffusion model as the world model to generate complete rollouts at once instead of auto-regressive generation.

[30] introduces the Q-score matching (QSM), which iteratively matches the parameterized score of a policy with the action gradient of its Q-function. Considering the low inference efficiency and long training time of diffusion models in RL training, [33] and [34] use consistency models instead of diffusion models and implement policy training under the actor-critic framework, achieving excellent performance in continuous control tasks.

## 2.2 Visual Reinforcement Learning

Compared to state-based RL, visual RL is faced with high-dimensional state space and continuous action space and is sensitive to training parameters and random seeds, which leads to unstable training and sample inefficiency. Image data augmentation [43–45] is a common technique to alleviate the above problems. In addition, auxilliary losses are usually combined to improve the efficiency of state representation learning from the image, such as contrastive learning loss [46], state representation loss [47, 48], action and state representation loss [49], and self-supervised loss [50]. Recent works have focused on enhancing the stability of visual RL from a micro perspective of neural networks. For example, [51] proposes the visual dead trial phenomenon and introduces an adaptive regularization method for convolutional features. [52] proposes the concept of dormant neuron phenomenon to explain the behavior of the policy network during RL training. [53] controls the dormant ratio of the policy network during training so that it achieves the SOTA performance on multiple tasks.

# 3 Preliminary

## 3.1 Reinforcement Learning

Online RL solves sequential decision problems, typically modeled through Markov Decision Processes (MDP). MDP is represented by 6 tuples $(\mathcal{S}, \mathcal{A}, \mathcal{R}, \mathcal{T}, \rho_0, \gamma)$. Here, $\mathcal{S}$ is the state space, $\mathcal{A}$ is the action space, $\mathcal{R}$ and $\mathcal{T}$ represent the reward function and state transition function of the environment, respectively. $\rho_0$ is the initial distribution of the state, and $\gamma$ is the discount factor. In visual RL, it is difficult for agents to directly obtain the state $s_t$ from the image $o_t \in \mathcal{O}$, where $\mathcal{O}$ is observation space. Therefore, an image encoder $f(\cdot)$ is usually required, and the state is estimated from the image through this encoder. The goal of the agent is to learn an optimal policy $\pi^*$ and the corresponding encoder $f^*$ to maximize the expected cumulative reward $\mathbb{E}_{\pi(f(\cdot))}[\sum_{t=0}^{\infty} \gamma^t r_t]$ under that policy.

## 3.2 Consistency Policy

The consistency model [32] is a new diffusion model proposed to address the time inefficiency caused by hundreds of reverse diffusion steps in diffusion models. It replaces the iterative denoising process with learned score functions in traditional diffusion models by constructing a mapping between noise and denoised samples, and directly maps any point on the probability flow ordinary differential equation (ODE) trajectory to the original data in the reverse diffusion process. Thus, it only requires a small number of steps or even one step to achieve the generation from noise to denoised data. Consistency policy [33, 34] is a new policy representation under the actor-critic framework, which replaces traditional Gaussian models with the consistency model and updates the policy by maximizing the state-action value. Consistency policy is defined as

$$\pi_\theta(a_t|s_t) \triangleq c_{skip}(\tau_k)a_t^{\tau_k} + c_{out}(\tau_k)F_\theta(a_t^{\tau_k}, \tau_k|s_t) \tag{1}$$

where $\{\tau_k|k \in [N]\}$ a sub-sequence of time points on the time period $[\epsilon, K]$ with $\tau_1 = \epsilon$ and $\tau_N = K$. $a_t^{\tau_k}$ is the noised action and $a_t^{\tau_k} = a_t + \tau_k z$ where $z \sim \mathcal{N}(0, I)$ is Gaussian noise. $F_\theta$ is a trainable network that takes the state $s_t$ as a condition and outputs an action of the same dimension as the input $a_t^k$. $c_{skip}(\cdot)$ and $c_{out}(\cdot)$ are differentiable functions such that $c_{skip}(\epsilon) = 1$ and $c_{out}(\epsilon) = 0$ to ensure consistency policy is differentiable at $\tau_k = \epsilon$. $\epsilon$ is a real number close to 0. To train this policy, [33] directly applies the above policy to the actor-critic framework and updates the policy using the following the Q-loss, which is named Consistency-AC.

$$\mathcal{L}_a(\theta) = -\mathbb{E}_{s_t \sim \mathcal{B}, a_t \sim \pi_\theta}[Q_\phi(s_t, a_t)] \tag{2}$$

where $\mathcal{B}$ is the replay buffer and $Q_\phi$ is the state-action value function. Compared to diffusion-based policies, consistency policy have significant advantages in inference speed and performance in online learning tasks [33].

### 3.3 Dormant Ratio of Neural Networks

The expressive ability is crucial for training the policy with RL. [29] proposes the concept of dormant ratio $\beta_r$, which quantifies the expression ability of a neural network by calculating the proportion of dormant neurons in the neural networks.

$$\beta_r = \frac{\sum_l H_\tau^l}{\sum_l N^l} \tag{3}$$

where $N^l$ represents the number of neurons in the $l$-th layer. $H_\tau^l$ is the number of neurons in the $l$-th layer whose score $s_i^l$ is less than $\tau$. The score of each neuron is calculated as follows:

$$s_i^l = \frac{\mathbb{E}_{x \in \mathcal{D}}|h_i^l(x)|}{\frac{1}{N^l}\sum_{k \in l}\mathbb{E}_{x \in \mathcal{D}}|h_k^l(x)|} \tag{4}$$

Here $h_i^l(\cdot)$ is the activation function of the $i$-th neuron in the $l$-th layer. $\mathcal{D}$ is the distribution of the input $x$. In the following sections of this paper, we use the dormant ratio to evaluate the expression ability of consistency policy during the training.

As introduced in [29], the dormant ratio of a neural network indicates the proportion of inactive neurons and is typically used to measure the activity of the network. A higher dormant ratio implies fewer active neurons in the network, implying the network's capacity and expressiveness are damaged. In RL, the episode return is closely related to the dormant ratio of the policy network. A higher dormant ratio results in more lazy action outputs, inactive agent behavior, and lower episode returns; conversely, when policy performance is good, the policy network is usually more active, and the dormant ratio is typically lower. This phenomenon has been reported in [29, 53–55].

## 4 Is Consistency-AC Applicable to Visual RL?

**Does the non-stationary distribution in online RL affect the training of consistency models?** Unlike offline RL, online RL does not have pre-collected datasets. The data distribution for training the policy is constantly changing with policy improvement. So, whether this non-stationarity distribution affects the training of consistency models is a question that needs to be explored. In order to investigate the impact of non-stationarity of data for consistency model training, we record the dormant ratio of the policy network during consistency model training under two different settings: online training and offline training. We employ two tasks (MuJoCo Halfcheetah and MuJoCo Walker2d) and conduct 4 random seeds for each setting. In order to eliminate the impact of Q-loss, we follow the behavior clone setting and train the consistency model with consistency loss [32] using data from offline datasets or online replay buffers. The distribution of the data in the replay buffer varies with policy improvement. The results are shown in Figure 1. Although there is a difference in the dormant ratios between online and offline learning settings in the Halfcheetah task, the overall trend is the same. We speculate that this difference is caused by the diversity of the samples included in the dataset. For the Walker2d task, the dormant ratios are nearly the same under two different settings. Therefore, we can infer that the non-stationary distribution of online RL does not significantly affect the consistency model training.

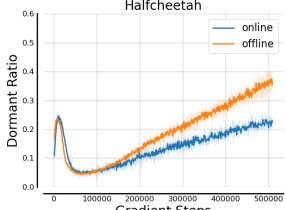

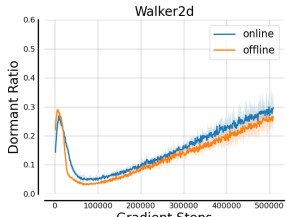

Figure 1: The dormant ratios of the policy under the online and offline training.

**Is the actor-critic framework suitable for training consistency policy?** The actor-critic framework is a highly effective policy training framework for online RL, in which the policy network achieves policy improvement by maximizing the value function. Some works [33, 34] directly apply consistency models to this framework. Although they achieve good results in RL tasks with low dimensional state spaces, whether the actor-critic training framework is compatible with consistency model training remains a question that needs further investigation. To evaluate the impact of the actor-critic framework on the training of consistency models, we compare the dormant ratios of policy

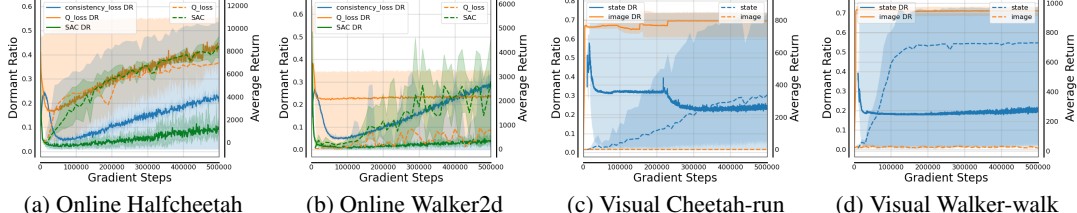

| (a) Online Halfcheetah | (b) Online Walker2d | (c) Visual Cheetah-run | (d) Visual Walker-walk |

Figure 2: The dormant ratios of the policy networks with different losses and observations.

networks under the consistency loss and Q-loss settings under the actor-critic framework. The results are shown in (a) and (b) of Figure 2, the solid line shows the dormant ratio of the policy while the dashed line shows the performance of the policy. When training the policy with the consistency loss, the dormant ratios of the network show a trend of first decreasing and then increasing. This means that the policy learns the distribution from the data and then overfits the distribution. When training the policy with the Q-loss, the dormant ratio of the policy network will rapidly increase and maintain a high value, which means that the policy network will quickly fall into local optima, making the policy no longer change. In addition, we can also see that when using the Q-loss, the variance of the dormant ratios is relatively large under different random seeds. When the dormant ratio is low, the policy network can iterate properly to learn good policy. Therefore, we can determine that the Q-loss under the actor-critic framework will destabilize the consistency policy training.

**Will high-dimensional state space exacerbate the degradation phenomenon of consistency policy?** Compared to RL with low dimensional state space, training stability in visual RL is still a challenge. In order to investigate whether the degradation phenomenon of consistency policy will become more significant under visual RL tasks, we compare the dormant ratios of the policy networks with the state as input and image as input on 2 tasks (Walker-walk and Cheetah-run in DeepMind control suite) under the setting of online learning. During the training process, only the Q-loss was used. To maintain consistency in the settings, we only count the dormant ratio of the multilayer perceptron (MLP) of the policy newtork in the image-based settings. The results are shown in (c) and (d) of Figure 2. Similar to using the state as input, in visual RL with the image as input, most of the neurons in the MLP of the policy network go dormant. Unlike the high variance of the former, the dormant ratios of consistency policy network in visual RL maintain a low variance and a high value. This indicates that there have been almost no successful trials under different random seeds. Therefore, we can infer that visual RL will exacerbate the instability of consistency policy training caused by the Q-loss under the actor-critic framework.

## 5 Consistency Policy with Prioritized Proximal Experience Regularization

**Consistency Policy with Entropy Regularization.** To solve the problem of consistency policy quickly falling into local optima under the influence of the Q-loss, we introduce policy regularization to stabilize policy improvement. Here, we employ entropy regularization to constrain policy behavior. The objective of RL is:

$$J(\theta) = \mathbb{E}_{s_t \sim \mathcal{B}, a_t \sim \pi_\theta} \big[ \sum_{t=0}^{\infty} \gamma^t r_t(s_t, a_t) \big] - \eta \mathbb{E}_{s_t \sim \mathcal{B}, a_t \sim \pi_\beta} [\log \pi_\theta(a_t|s_t)] \tag{5}$$

where $\pi_\beta$ is the proxy distribution required for policy regularization. Entropy regularization is a commonly method for stabilizing policy training in RL. When $\pi_\beta$ is a uniform distribution, the above objective is equal to maximum entropy RL, which maximizes the entropy of the policy while optimizing the return. The prerequisite for this method is to obtain the closed form of the policy distribution to calculate its entropy. However, for diffusion models or consistency models, obtaining the closed form of the policy distribution is very difficult. Thanks to the development of generative models, we can use score matching instead of solving analytic entropy in entropy regularization RL, thus achieving sample-based policy regularization. Therefore, the training loss of consistency policy with the entropy regularization is:

$$\mathcal{L}_a^r(\theta) = -\mathbb{E}_{s_t \sim \mathcal{B}, a_t \sim \pi_\theta} [Q_\phi(s_t, a_t)] + \eta \mathcal{L}_c(\theta) \tag{6}$$

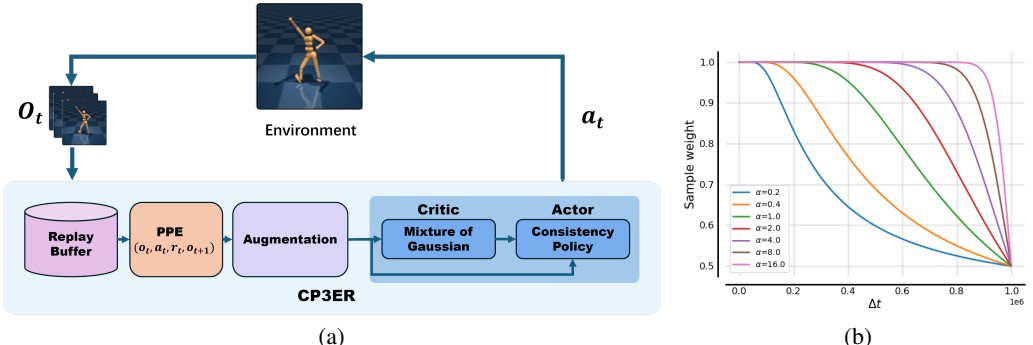

(a)                                                           (b)

Figure 3: (a) The framework of CP3ER, where PPE is the abbreviation of prioritized proximal experience. (b) The sampling weights $\beta$ with different $\alpha$.

where $\mathcal{L}_c$ is consistency loss defined by following:

$$\mathcal{L}_c(\theta) = \mathbb{E}_{k\sim\mathcal{U}(1,N-1),s_t\sim\mathcal{B},a_t\sim\pi_\beta,z\sim\mathcal{N}(0,I)}[\lambda(\tau_k)d(\pi_\theta(s_t,a_t^{\tau_{k+1}},\tau_{k+1}),\pi_{\bar{\theta}}(s_t,a_t^{\tau_k},\tau_k)) \quad (7)$$

Here $\lambda(\cdot)$ is a step-dependent weight function, $d(\cdot,\cdot)$ is a distance metric. Since there is no need to obtain the closed form of the proxy distribution, only the data under that distribution needs to be obtained, making the selection of proxy distribution flexible. The remaining question is how to construct a suitable proxy distribution $\pi_\beta$.

**Prioritized Proximal Experience Regularization.** When the proxy distribution is uniform, this method approximates the maximum entropy consistency policy (MaxEnt CP). It should be noted that the difference between the proxy distribution and the optimal policy distribution can lead to difficulty in optimizing the above objectives. When the proxy distribution is far from the optimal policy or the proxy distribution is complex, the above optimization objectives require more samples to converge to better results. To better balance training stability and sample efficiency, we propose the prioritized proximal experience regularization (PPER). Specifically, when sampling data from the replay buffer, we design sampling weight $\beta$ for each data instead of sampling the data uniformly.

$$\beta = \frac{1}{1 + \exp\left(2\alpha - \alpha\frac{2|\mathbf{B}|}{\Delta t}\right)} \quad (8)$$

where $\alpha$ is the hyperparameter and $\Delta t$ is the interval between the sample generation step and the current step. $|\mathbf{B}|$ is the capacity of the replay buffer. The curves with different $\alpha$ are shown in Figure 3 (b). In the above settings, data closer to the current step will be sampled with a higher probability, while data farther away will have a lower probability. We refer to the above sampling method as a prioritized proximal experience (PPE).

For policy evaluation, we suggest a distributional value function instead of a deterministic value function to ensure the stability and accuracy of the value estimation. Precisely, we follow [56] and use a mixture of Gaussian (MoG) to model the distribution of the state-action value. When MoG is employed for value distribution, the following loss is used to update Q-function.

$$\mathcal{L}_q(\phi) = -\mathbb{E}_{(s_t,s_{t+1})\sim\mathcal{B},a_t\sim\pi_\theta}\left[\frac{1}{M}\sum_{i=1}^{M}\log Z_\phi^{(s_t,a_t)}(r_t(s_t,a_t)+\gamma z_i')\right], \text{where} \begin{cases} z_i' \sim Z_{\bar{\phi}}^{(s_{t+1},a_{t+1})} \\ a_{t+1} \sim \pi_\theta(s_{t+1}) \end{cases} \quad (9)$$

where $Z_\phi^{(s_t,a_t)}$ is the estimated value distribution. According to the equation (9), we need to sample $M$ target Q-values $z_i'$ and update the value distribution. Different from [56], we sample only one next action $a_{t+1}$ instead of multiple actions to reduce the time cost and find that this simplification can achieve good experiment results. Considering the simplicity and efficiency of DrQ-v2 [57], our proposed consistency policy with prioritized proximal experience regularization (CP3ER) is built based on DrQ-v2. The framework is shown as Figure 3 (a). We sample the data from the replay buffer, and augment the image with the techniques in DrQ-v2. Thanks to the natural randomness of the action from consistency policy, our method no longer requires additional exploration strategies.

In addition, we only used a single Q-network to estimate the mean and variance of the mixture of Gaussian instead of double Q-network. We consider prioritized proximal experience regularization when updating consistency policy and used equation (9) when training the Q-network, which differs from DrQ-v2. The complete algorithm is included in the appendix B.1.

# 6 Experiments

In this section, we evaluate the proposed method from the following aspects: 1) Does CP3ER have performance advantages compared to current SOTA methods? 2) Can policy regularization improve the behavior of the policy? 3) What is the impact of different modules on the performance?

## 6.1 Visual Continuous Control Tasks

**Environment Setup.** We evaluate the methods on 21 visual control tasks from DeepMind control suite [58] and Meta-world [59]. We split these tasks into three domains, including 8 medium-level tasks in the DeepMind control suite, 7 hard-level tasks in the DeepMind control suite, and 6 tasks in the Meta-world. The details of each domain are included in the appendix C.

**Baselines.** We compare current advanced model-free visual RL methods, including DrQ-v2 [57], ALIX [60] and TACO [61]. The more detailed results are shown in the appendix C.

### 6.1.1 Does CP3ER have performance advantages compared to current SOTA methods?

**Medium-level tasks in DeepMind control suite.** We evaluate CP3ER on 8 medium-level tasks [57] in DeepMind control suite. The results are shown in Figure 4. From the left part of Figure 4, it can be seen that compared to the current SOTA methods, our proposed CP3ER has achieved better sample efficiency. It should be noted that TACO uses auxiliary losses of action and state representation during training to improve sample efficiency. Moreover, our proposed CP3ER uses no additional losses or exploration strategies. On the right part of Figure 4, we compare the mean, Interquartile Mean (IQM), median, and optimal gap of these methods. CP3ER has significant advantages in all metrics and has more minor variance. This means that CP3ER has better training stability.

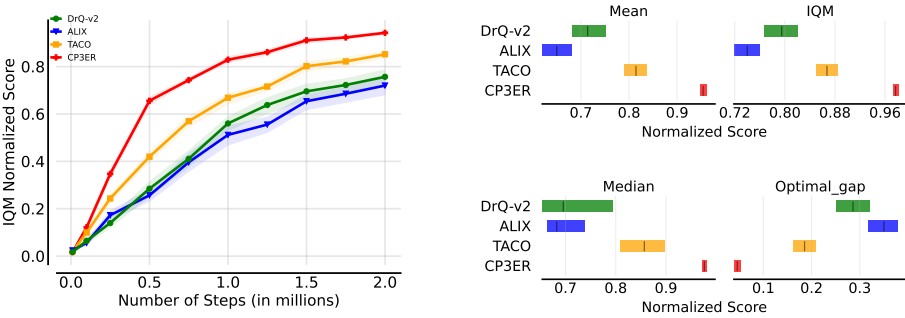

Figure 4: Results on medium-level tasks in DeepMind control suite with 5 random seeds.

**Hard-level tasks in DeepMind control suite.** We also evaluate CP3ER on 7 challenging tasks [53, 57] in the DeepMind control suite. It should be noted that all tasks here only train 5M frames, rather than the commonly used 30M frames in other work[53, 57]. This means it is a very hard challenge. From the results on the left part of Figure 5, it can be seen that most methods have yet to learn effective policy within 5M frames. Our proposed CP3ER surpasses the performance of all methods without relying on any additional loss or exploration strategies. Moreover, it has significant advantages on all metrics including mean, IQM, median, and optimal gap.

**Meta-world.** We also evaluated the methods on 6 complex tasks in the Meta-world. The results are shown in Figure 6. We record the success rates of the tasks, and all results are based on the success rates. Compared to other methods, CP3ER can quickly learn effective manipulation policy, and the success rate can reach nearly 100% in all tasks within 2M steps. By comparing the mean, IQM, median, and optimal gap of the success rates, CP3ER has a significant performance advantage with minimal variance.

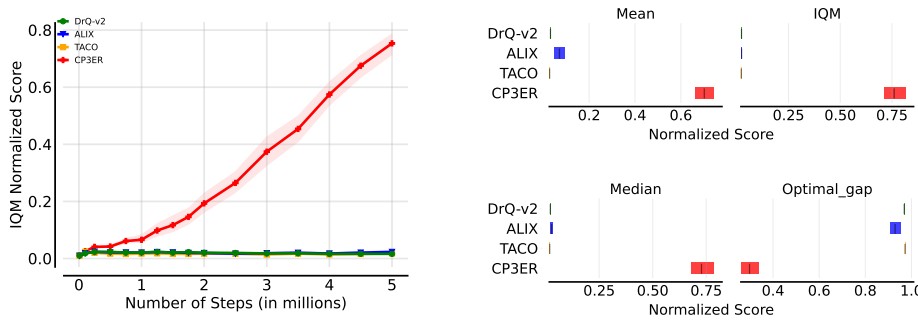

Figure 5: Results on hard-level tasks in DeepMind control suite with 5 random seeds.

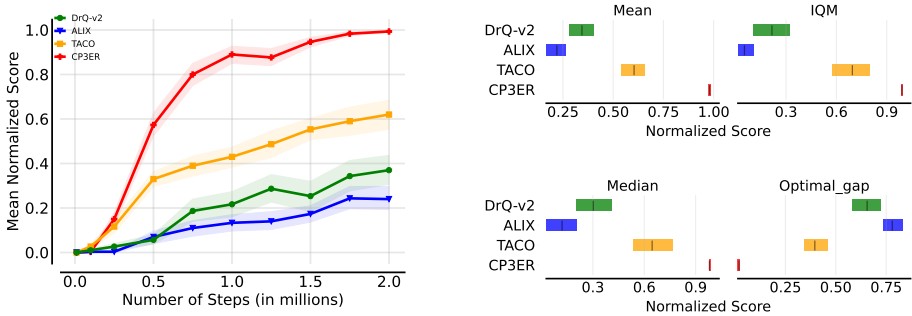

Figure 6: Results on Meta-world tasks with 5 random seeds.

## 6.2 Ablation Study

### 6.2.1 Can policy regularization improve the behavior of the policy during training?

**Action distribution analysis with toy example.** In order to further explore the impact of policy regularization on the training, we borrow the 1D continuous bandit problem [9] to analyze the policy behavior. The green curve in Figure 7 (a) shows the reward function. Within a narrow range of actions, the agent receives higher rewards, while within a broader range, the agent can only receive suboptimal rewards. Therefore, the policy needs strong exploration ability to achieve the highest return. We compare Gaussian policy with entropy regularization (MaxEnt GP), Consistency-AC[33] and consistency policy with entropy regularization (MaxEnt CP), and record the action distribution during the training. As shown in Figure 7, Consistency-AC quickly converges to the local optimal value with the Q-loss. Policy regularization ensures the diversity of action distribution during consistency policy training, preventing the policy from falling into local optima too early. Moreover, consistency policy has robust exploration compared to the Gaussian policy and achieves higher returns.

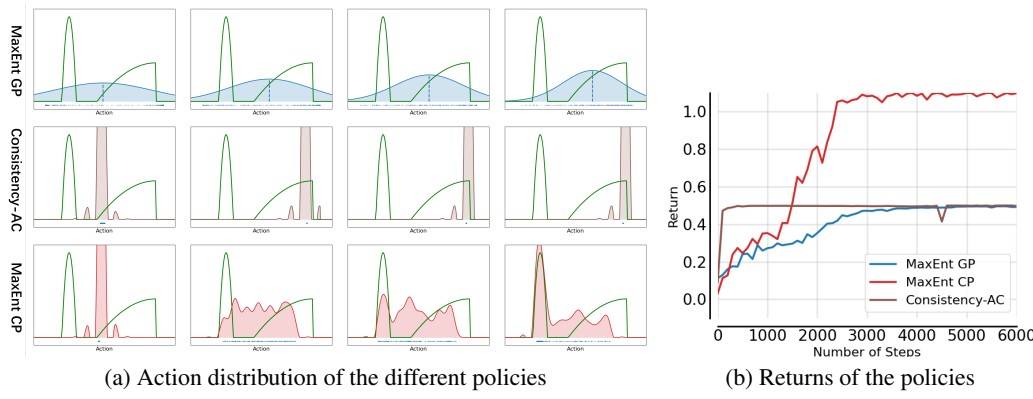

(a) Action distribution of the different policies     (b) Returns of the policies

Figure 7: Results on the toy example. Left part is action distributions during training, while right is returns of different policies.

**Dormant ratio analysis.** We have shown that the Q-loss can rapidly increase the dormant ratio of the consistency policy network, leading to a loss of policy diversity. In order to analyze whether entropy regularization can alleviate the phenomenon, we record the dormant ratios of the policy networks during the training in 3 tasks. The results are shown in Figure 8. Compared to the rapid increase in the dormant rate in Consistency-AC, CP3ER has a lower dormant ratio, which means that entropy regularization can effectively reduce the dormant ratios of consistency policy.

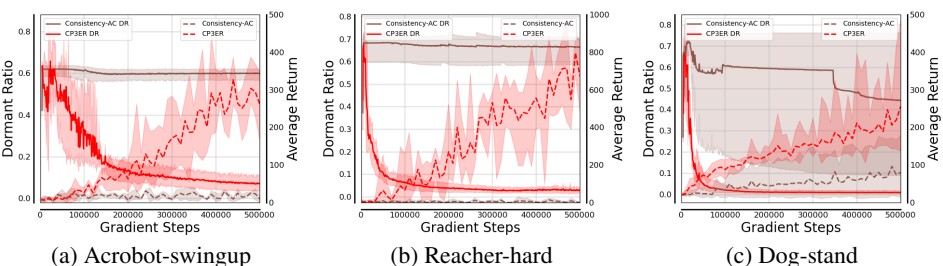

(a) Acrobot-swingup  (b) Reacher-hard  (c) Dog-stand

Figure 8: Dormant ratios of the policy networks on different tasks with 5 random seeds.

### 6.2.2 What is the impact of different modules on the performance?

We conduct ablation studies in 2 tasks to evaluate the contribution of each module in the proposed method. In addition, to analyze the impact of proxy distribution for policy regularization on performance, we also compare several candidates, including uniform distribution or behavioral policy in the replay buffer. The results are shown in Figure 9. It is noticeable that policy regularization is crucial for consistency policy. Without policy regularization, consistency policy (CP3ER w/o PPER) makes it difficult to learn meaningful behavior in the tasks. The proxy distribution also has an impact on the performance. Using uniform distribution to regularize policies can make the policy (CP3ER w. MaxEnt) improvement difficult, resulting in low sample efficiency. Compared to using behavior distribution in the replay buffer (CP3ER w. URB), the policy (CP3ER) obtained through prioritized proximal experience sampling has a closer distribution to the current policy, making policy optimization easier and resulting in higher sample efficiency. In addition, we find that the performance of CP3ER is significantly better than the baseline (DrQ-v2), indicating that the feasible usage of consistency policy can help solve visual RL tasks.

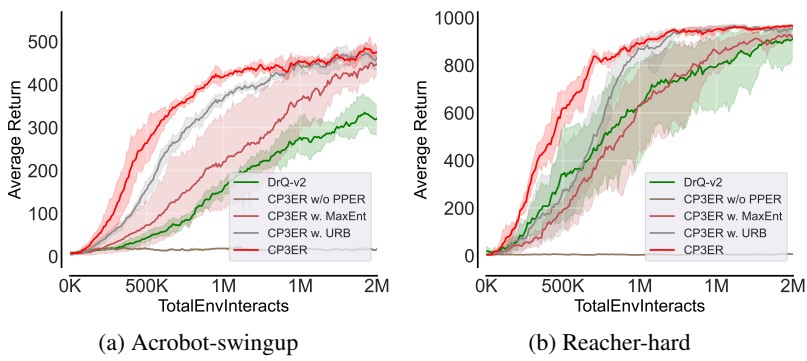

(a) Acrobot-swingup  (b) Reacher-hard

Figure 9: Results of ablation study on 2 visual control tasks with 4 random seeds.

## 7 Conclusion

In this paper, we analyze the problems faced by extending consistency policy to visual RL under the actor-critic framework and discover the phenomenon of the collapse of consistency policy during training under the actor-critic framework by analyzing the dormant ratio of the neural networks. To address this issue, we propose a consistency policy with prioritized proximal experience regularization (CP3ER) that effectively alleviates the training collapse problem of consistency policy. The method is evaluated on 21 visual control tasks and shows significantly better sample efficiency and performance

than the current SOTA methods. It is worth mentioning that, to the best of our knowledge, our proposed CP3ER is the first method to apply diffusion/consistency models to visual RL tasks.

Our experimental results show that the consistency policy benefits from its expressive ability and ease of sampling, effectively balancing exploration and exploitation in RL with high-dimensional state space and continuous action space. It achieves significant performance advantages without any auxiliary loss and additional exploration strategies. We believe that consistency policy will play an essential role in visual RL. There are still some issues worth exploring in future work. Firstly, auxiliary losses for representation in current visual RL have the potential to improve the performance of consistency policy. Secondly, the diversity of behavior in consistency policy is crucial for RL exploration. This paper only discusses the stability of policy training and does not analyze the diversity of behavior during training, which will help improve the exploration performance of policies. In addition, consistency policy under the on-policy framework is also a direction worth exploring.

## 8 Acknowledgments

This work is supported by the National Natural Science Foundation of China (NSFC) under Grants No. 62103409, No. 62136008, No. 62293545 and in part by the International Partnership Program of the Chinese Academy of Sciences under Grant 104GJHZ2022013GC.

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

# A    More Results on Dormant Ratios

To demonstrate the phenomenon of policy degradation in Consistency-AC, we conduct experiments with different settings on more tasks and analyze the effect on the dormant ratio. For analyzing the impact of non-stationary data distribution on consistency model training in online RL, we employ 3 classic tasks from D4RL. The results are shown in Figure 10. It can be seen that the non-stationary distribution caused by online training does not significantly affect the dormant ratio of the policy network.

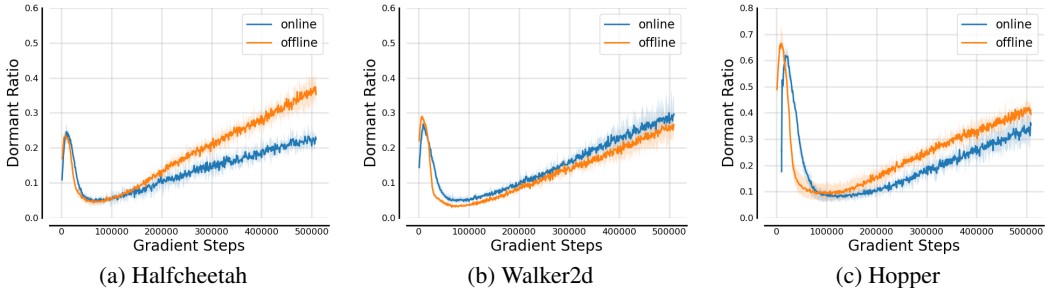

| (a) Halfcheetah | (b) Walker2d | (c) Hopper |

Figure 10: The dormant ratios of the policy under the online and offline training. All results are averaged over 4 random seeds, and the shaded region stands for standard deviation across different random seeds.

For analyzing the impact of the loss function and high-dimensional state input on the consistency policy, we conduct experiments on 5 tasks separately. Among them, the results of 2 tasks are presented in the main part, and the results of the remaining 3 tasks are shown here.

Figure 11 shows the impact of different loss functions on the performance and the dormant ratio of consistency policy. Compared to SAC policy, consistency policy with Q-loss achieves the higher dormant ratio and the worse performance. Especially on the Finger-turn-hard task, the policy under the Consistency-AC framework hardly learn any meaningful behavior, and the dormant ratios remain at a high value. Figure 12 shows the impact of different observation inputs on Consistency-AC. Compared to state-based settings, image-based policy is more difficult to learn meaningful behavior, and its dormant ratio also maintains a higher value.

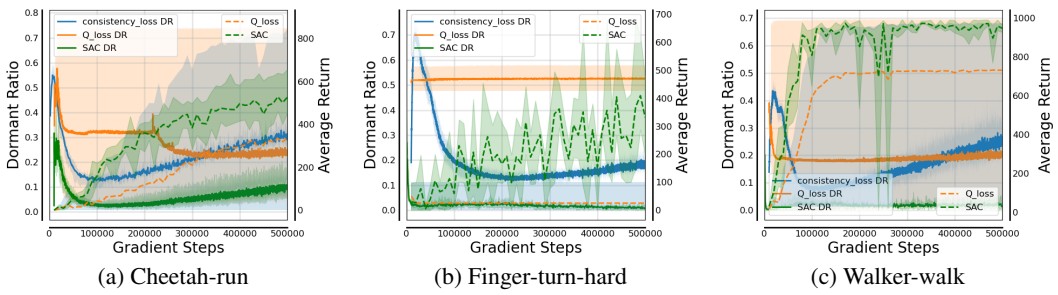

| (a) Cheetah-run | (b) Finger-turn-hard | (c) Walker-walk |

Figure 11: The dormant ratios of the policy with different training loss. All results are averaged over 4 random seeds, and the shaded region stands for standard deviation across different random seeds.

# B    Implementation Details

In this paper, we propose Consistency Policy with Prioritized Proximal Experience Regularization (CP3ER), which is built on the basis of DrQ-v2 and its framework is shown in Figure 13. Compared to DrQ-v2, its difference lies in the use of Prioritized Proximal Experience (PPE) when sampling data from the replay buffer, and the use of consistency policy instead of Gaussian policy in the

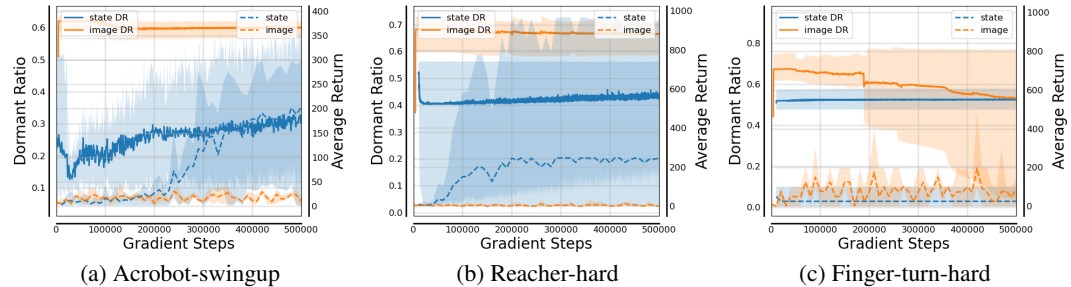

| (a) Acrobot-swingup | (b) Reacher-hard | (c) Finger-turn-hard |

Figure 12: The dormant ratios of the policy with different observations. All results are averaged over 4 random seeds, and the shaded region stands for standard deviation across different random seeds.

actor. In addition, it employs a mixture of Gaussians to model the value distribution rather than the deterministic double Q-networks in DrQ-v2.

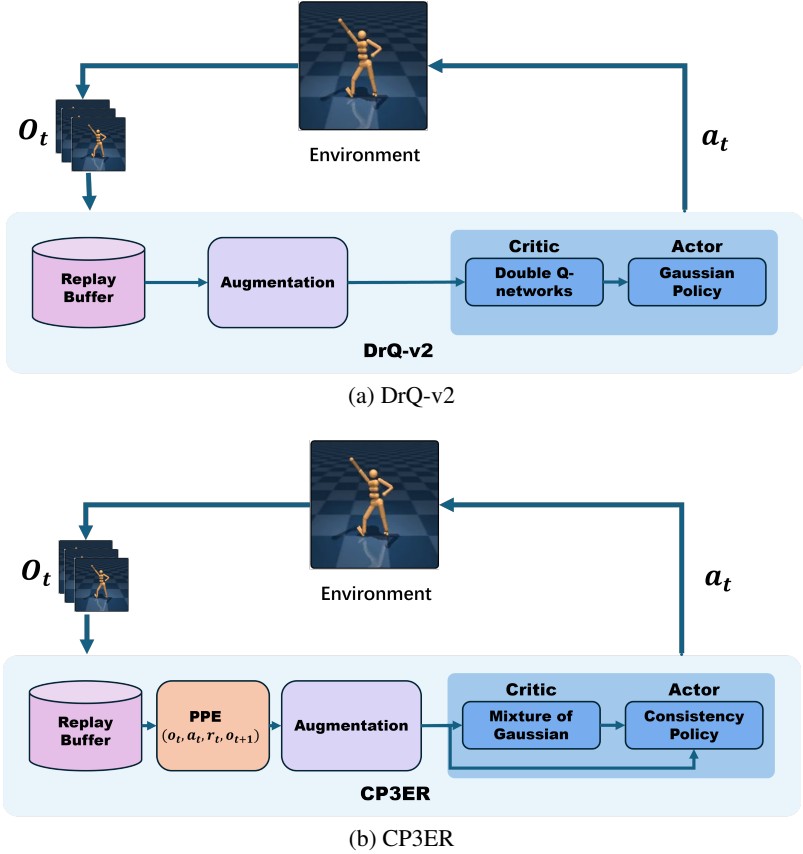

(a) DrQ-v2

(b) CP3ER

Figure 13: Comparison between DrQ-v2 and CP3ER.

## B.1 Procedure of the proposed algorithm

We have demonstrated the complete procedure of CP3ER in Algorithm 1, 2 and 3.

## B.2 Hyperparameters

We present a summary of all the hyperparameters for CP3ER in Table 1, where DMC is the abbreviation of DeepMind control suite. It is worth noting that for tasks in different domains, only the

**Algorithm 1** Algorithm of CP3ER

**Input:**
$f_\xi, \pi_\theta, Q_\phi$: parametric networks for encoder, policy and Q-function respectively.
aug: random shifts image augmentation.
$T, B, \alpha, \tau$: training steps, mini-batch size, learning rate, target update rate.
**Training routine:**
    **for** each timestep $t = 1..T$ **do**
        $a_t \leftarrow \pi_\theta(f_\xi(o_t))$
        $o_{t+1} \sim P(\cdot|o_t, a_t)$
        $\mathcal{B} \leftarrow \mathcal{B} \cup (o_t, a_t, R(o_t, a_t), o_{t+1})$
        $(o_t, a_t, r_t, o_{t+1}) \sim \mathcal{B}$ with PPE sampling method
        UPDATECRITIC$(o_t, a_t, r_t, o_{t+1})$
        UPDATEACTOR$(o_t, a_t)$
    **end for**

---

**Algorithm 2** Training critic

    **procedure** UPDATECRITIC$(o_t, a_t, r_t, o_{t+1})$
        $h_t, h_{t+1} \leftarrow f_\xi(aug(o_t)), f_\xi(aug(o_{t+1}))$
        $a_{t+1} \leftarrow \pi_\theta(o_{t+1})$
        Compute $\mathcal{L}_q(\phi)$ using Equation (9)
        $\xi \leftarrow \xi - \alpha \nabla_\xi \mathcal{L}_{q,\xi}(\phi)$
        $\phi \leftarrow \phi - \alpha \nabla_\phi \mathcal{L}_{q,\xi}(\phi)$
        $\bar{\phi} \leftarrow (1 - \tau)\bar{\phi} + \tau\bar{\phi}$
    **end procedure**

---

learning rate and feature dimension are different, while other parameters are the same for all tasks. The parameters of our proposed method are not task-sensitive, which helps it be applied to a wider range of visual control tasks without the need for fine-tuning of parameters.

# C   More Results

In this section, we present more detailed experimental results on tasks in DeepMind control suite and Meta-world, including performance curves during the training, performance profiles, and probability of performance improvement. We compare CP3ER to 4 baselines including DrQ-v2 [57], ALIX [60], TACO [61] and DrM [53]. All evaluations are based on a single NVIDIA GeForce RTX 2080 Ti. For CP3ER, training a run with 2M frames on this device will take about 13 hours.

- DrQ-v2: `https://github.com/facebookresearch/drqv2`
- ALIX: `https://github.com/Aladoro/Stabilizing-Off-Policy-RL`
- TACO: `https://github.com/FrankZheng2022/TACO`
- DrM: `https://github.com/XuGW-Kevin/DrM`

---

**Algorithm 3** Training actor

    **procedure** UPDATEACTOR$(o_t, a_t)$
        $h_t \leftarrow f_\xi(o_t)$
        $\hat{a}_t \leftarrow \pi_\theta(o_t)$
        Compute $\mathcal{L}_a^r(\theta)$ using Equation (6)
        $\theta \leftarrow \theta - \alpha \nabla_\theta \mathcal{L}_a^r(\theta)$
        $\bar{\theta} \leftarrow (1 - \tau)\bar{\theta} + \tau\bar{\theta}$
    **end procedure**

| Parameter | Setting |
|-----------|---------|
| Replay buffer capacity | $10^6$ |
| Action repeat | 2 |
| Seed frames | 4000 |
| Exploration steps | 10000 |
| $n$-step returns | 3 |
| Mini-batch size | 256 |
| Discount $\gamma$ | 0.99 |
| Optimizer | Adam |
| Learning rate | $8 \times 10^{-5}$ (Hard-level tasks in DMC) |
| | $10^{-4}$ (Medium-level tasks in DMC & Meta-world) |
| Soft update rate | 0.01 |
| Features dimension | 100 (Hard-level tasks in DMC) |
| | 50 (Medium-level tasks in DMC & Meta-world) |
| Hidden dimension | 1024 |
| Number of Gaussian mixtures for Critic | 3 |
| Number of samples $M$ for update Critic | 20 |
| Parameter $\alpha$ for PPE | 2.0 |
| Coefficient for consistency loss $\eta$ | 0.05 |
| $\tau$-Dormant ratio $\eta$ | 0.025 |

Table 1: The hyper-parameters for CP3ER.

## C.1  Results on Medium-level Tasks in DeepMind Control Suite

In this subsection, we show the detail results on the 8 medium-level tasks [57] in DeepMind control suite. These tasks include: acrobot-swingup, cheetah-run, finger-turn-hard, hopper-hop, quadruped-run, quadruped-walk, reacher-hard and walker-walk. From the results in Figure 14, it can be seen that our proposed CP3ER has better sample efficiency and performance on almost all tasks, even though it does not use any loss for representation learning. In addition to the performance curve during the training, we also demonstrate the performance profiles at different checkpoints and the probability of performance improvement. According to the results in Figure 15, CP3ER is significantly superior to other baseline methods.

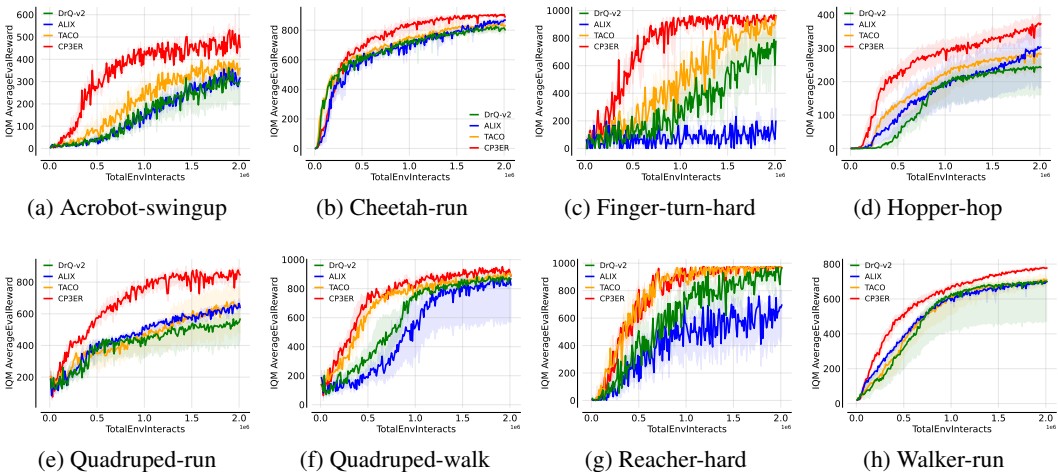

Figure 14: Performance of CP3ER against baseline algorithms DrQ-v2, ALIX, and TACO on the medium-level tasks in DeepMind control suite. All results are averaged over 5 random seeds, and the shaded region stands for standard deviation across different random seeds.

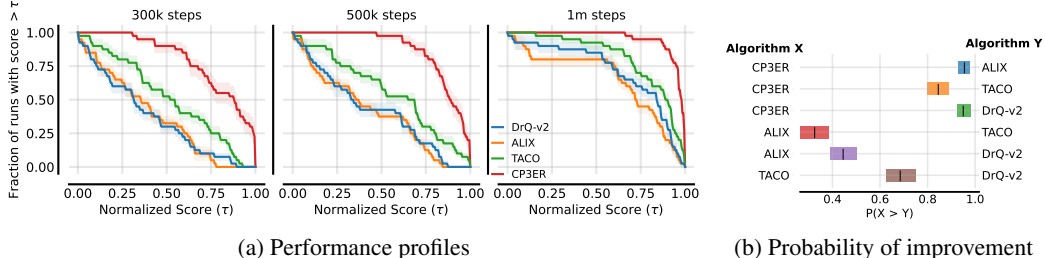

(a) Performance profiles           (b) Probability of improvement

Figure 15: Performance profiles and probabilities of improvement of different methods.

## C.2    Results on Hard-level Tasks in DeepMind Control Suite

In this subsection, we show the detail results on the 7 hard-level tasks [53, 57] in DeepMind control suite. These tasks include: dog-run, dog-stand, dog-trot, dog-walk, humanoid-run, humanoid-stand and humanoid-walk. In addition to the 3 comparison baselines considered in the previous subsection, we also consider DrM which achieves SOTA performance on hard-level tasks in DeepMind control suite by weight perturbation and exploration strategies. Figure 16 shows the results CP3ER against several baselines. It is difficult for DrQ-v2, ALIX, and TACO to learn meaningful policy within the 5M framework on these tasks. CP3ER achieves comparable performance to DrM on 4 tasks: dog-run, dog-stand, dog-trot, and dog-walk without any exploration strategy. In addition, CP3ER significantly outperforms DrM in another 3 more challenging tasks. From the results in Figure 16, CP3ER also significantly outperforms other baselines on the tasks. Compared to the SOTA method DrM for difficult tasks, CP3ER also has a performance improvement probability of over 70%.

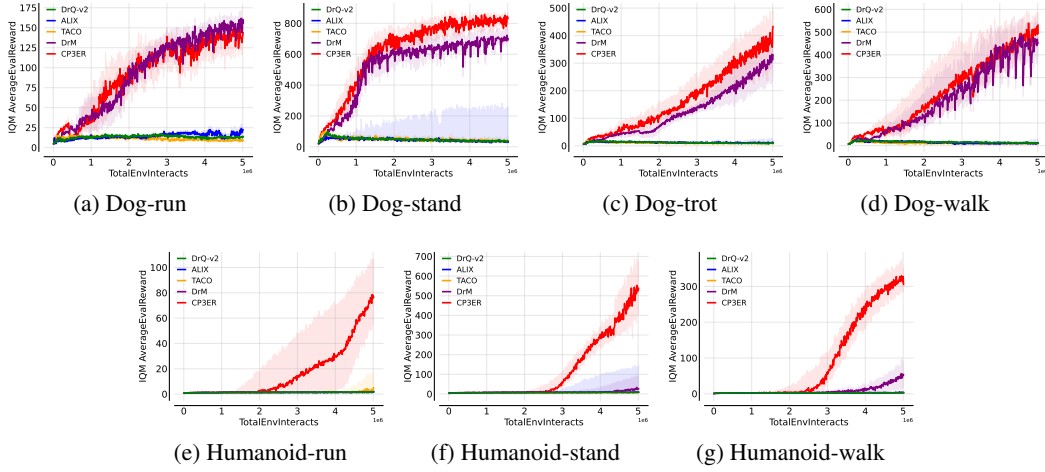

(a) Dog-run      (b) Dog-stand      (c) Dog-trot      (d) Dog-walk

(e) Humanoid-run      (f) Humanoid-stand      (g) Humanoid-walk

Figure 16: Performance of CP3ER against baseline algorithms DrQ-v2, ALIX, TACO and DrM on hard-level tasks in DeepMind control suite. All results are averaged over 5 random seeds, and the shaded region stands for standard deviation across different random seeds.

## C.3    Results on Tasks in Meta-world

In this subsection, we show the detail results on the 6 hard tasks [53] in Metat-world. These tasks include: assembly, disassemble, hammer, hand insert, pick place wall and stick pull. In these tasks, we compared 4 baselines, including DrM. It should be noted that in [53], DrM achieves SOTA performance on these hard tasks in Meta-world, but we cannot reproduce the corresponding performance based on the official codes. Therefore, this result is used for reference and may cannot represent the true performance of DrM. According to the results in Figure 18 and Figure 19, CP3ER achieves SOTA performance on all 6 hard-level tasks in Meta-world, and compared to the baselines, CP3ER has a significant performance advantage.

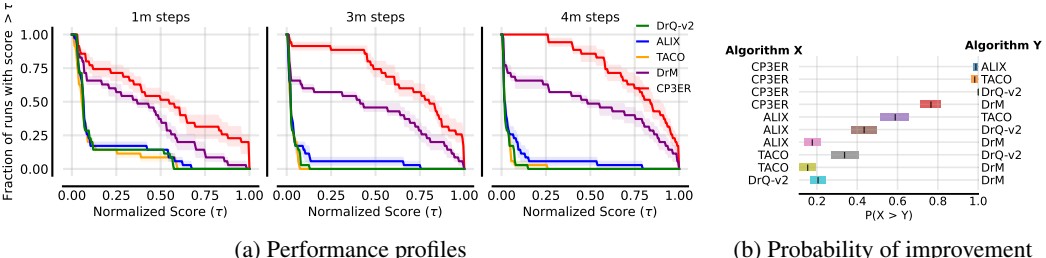

(a) Performance profiles

(b) Probability of improvement

Figure 17: Performance profiles and probabilities of improvement of different methods.

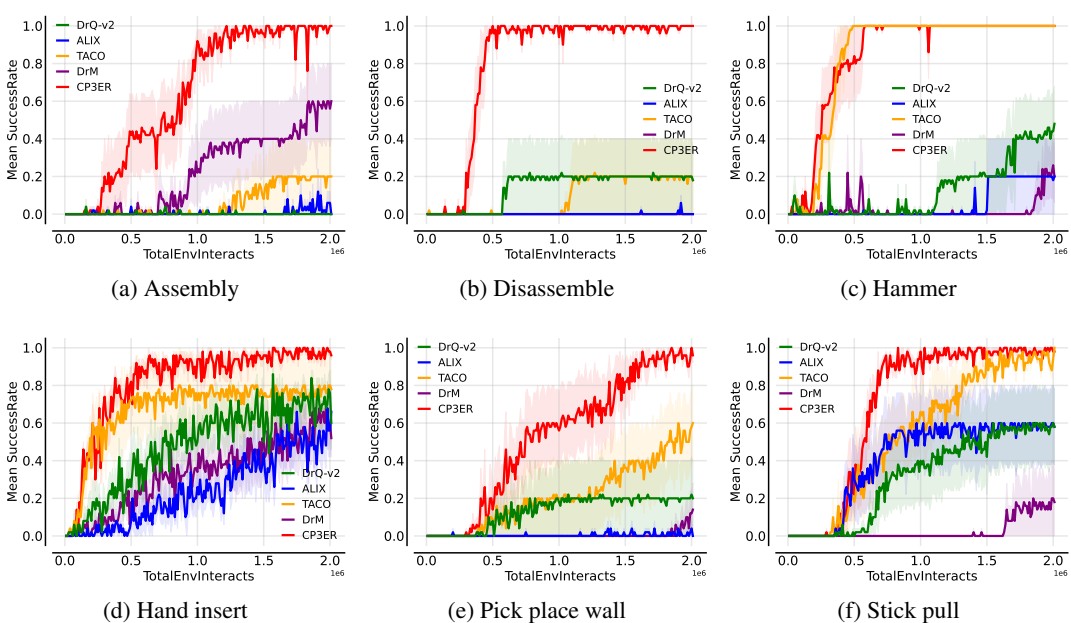

(a) Assembly

(b) Disassemble

(c) Hammer

(d) Hand insert

(e) Pick place wall

(f) Stick pull

Figure 18: Performance of CP3ER against baseline algorithms DrQ-v2, ALIX, TACO and DrM on tasks in Meta-world. All results are averaged over 5 random seeds, and the shaded region stands for standard deviation across different random seeds.

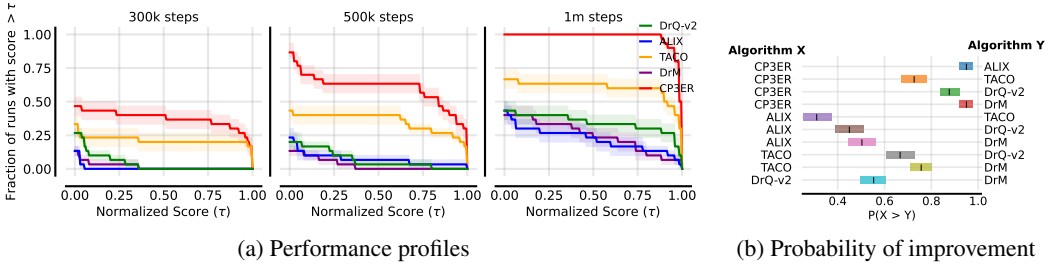

(a) Performance profiles

(b) Probability of improvement

Figure 19: Performance profiles and probabilities of improvement of different methods.

## C.4    Results on State-based Tasks

Sample efficiency has always been a highly concerned issue in visual RL, and high expressive power of the policy can improve exploration efficiency and thus increase sample efficiency. After introducing the diffusion model into visual RL, we find a serious phenomenon of policy degradation, which results in disastrous performance. Therefore, we propose CP3ER to stabilize the training of consistent policy. In addition to visual RL tasks, CP3ER theoretically has the potential to be applied to state-based RL tasks. According to the experimental setup in [34], we conduct a comparison to

state-based RL SOTA methods, and the experimental results are shown in the Table 2. It can be seen that compared to the current SOTA methods, CP3ER also has significant advantages in the tasks, which also proves the generalization of CP3ER on different observation tasks.

| Tasks | TD3 | SAC | PPO | MPO | DMPO | D4PG | DreamerV3 | CPQL | CP3ER |
|---|---|---|---|---|---|---|---|---|---|
| Acrobot Swingup | 46.8 | 33.2 | 34.4 | 80.6 | 98.5 | 125.5 | 154.5 | 183.1 | **362.9** |
| Finger Turn Easy | 337.6 | 371.4 | 275.2 | 430.4 | 593.8 | 524.5 | 745.4 | **874.1** | 867.45 |
| Finger Turn Hard | 334.4 | 344.8 | 5.06 | 250.8 | 384.5 | 379.2 | 841.0 | 864.6 | **912.65** |
| Hopper Hop | 40.0 | 41.7 | 0.0 | 37.5 | 71.5 | 67.5 | 111.0 | 130.1 | **299.3** |
| Hopper Stand | 322.7 | 270.9 | 2.2 | 279.3 | 519.5 | 755.4 | 573.2 | 902.1 | **904.1** |
| Walker Run | 274.5 | 445.9 | 131.7 | 539.5 | 462.9 | 593.1 | 632.7 | **683.8** | 646.5 |
| Average | 226 | 251.3 | 74.8 | 269.7 | 355.1 | 407.5 | 509.6 | 606.3 | **665.5** |

Table 2: Comparison of CP3ER and other methods on state-based RL tasks in DeepMind control suite.

In addition, we also compare the existing methods based on diffusion/consistency models following the settings in [33], and the results are shown in Table 3. It can be seen that compared to existing methods based on diffusion/consistency models, our method CP3ER, although aimed at visual RL problems, still has significant advantages when extended to state-based RL tasks.

| Online tasks | Diffusion-QL | Consistency-AC | CP3ER |
|---|---|---|---|
| Halfcheetah-m | 5745.9$\pm$388.5 | 6725.2$\pm$944.4 | **10699.1**$\pm$1054.7 |
| Hopper-m | **3675.5**$\pm$47.7 | 3589.7$\pm$163.4 | 3309.6$\pm$133.2 |
| Walker2d-m | 4316.2$\pm$612.1 | 3790.9$\pm$1677.5 | **5201.0**$\pm$111.3 |
| Average | 4579.2 | 4701.9 | **6403.2** |

Table 3: Comparison of CP3ER with diffusion/consistency based RL methods.

# D   Limitations

Although CP3ER enhances the training stability of consistency policy in the actor-critic framework and achieves excellent performance in visual control tasks, the policy diversity of CP3ER has not been thoroughly explored. In CP3ER, this diversity only depends on the multimodal actions in the replay buffer, which gradually disappears as the training steps increases. Therefore, CP3ER may face the risk of losing diversity in consistent policy, thereby weakening its exploration ability to a certain extent. In addition, there is a lack of theoretical analysis on CP3ER. Although it is based on the actor-critic framework, its policy improvement and convergence property require more rigorous theoretical analysis.

# E   Broader Impacts

This work mainly focuses on the field of visual RL and proposes a new method that may significantly improve the efficiency of visual RL. This method may improve the efficiency of robot skill learning and have a wide impact on visual control fields such as robots, but it will not involve ethical and safety issues. Therefore, this work should not have negative social impacts.

