# OpenReview forum: "Generalizing Consistency Policy to Visual RL with Prioritized Proximal Experience Regularization"
_NeurIPS.cc/2024/Conference — NeurIPS 2024 poster_

### Official Review · Reviewer_ceLx · 2024-07-08

**Soundness:** 3
**Presentation:** 2
**Contribution:** 3
**Rating:** 5
**Confidence:** 4

**Summary:**

The paper presents a novel method using consistency models as a policy parameterization to address online reinforcement learning. The paper first uses the dormant ratio metric to study properties of the consistency-AC method and determine that it is suitable for online RL. Then, it presents the CP3ER method, which adds an entropy regularization term to consistency AC in order to make it suitable for online RL. Finally the paper presents results showing strong performance in image-based RL benchmarks.

**Strengths:**

- Diffusion and consistency policies are an increasingly important research area, and in particular there has been limited work training these with online reinforcement learning. This paper seems to take an important step towards that direction, and thus has the potential to be significant.
 - Although several other works have used similar setups to apply DP/CP to offline RL, the specific formulation the authors employ to address online RL is novel as far as I can tell.
 - The method overall seems principled and the results presented in the paper seem promising.
 - Although there are issues with section 4 which I discuss extensively in the 'Questions' section, I like what the authors are trying to do and think if the section is written by more detail it can strengthen the case for their method.

**Weaknesses:**

- I think the biggest issue with the paper is that the story is a bit muddled. The core contribution of the paper seems to be CP3ER and the PPER regularization method. These things should be able to work for both low-dimensional and image-based observations. However, the authors specifically state that their goal is to address image-based observations. Why? Does this algorithm no longer show any benefit in these settings? The algorithm doesn't have any contributions that specifically focus on addressing image-based observations so I don't understand why that is a central focus of the evaluation.
    - In order to properly contextualize this algorithm I think it would be necessary to compare it to low-dim baselines like SAC and PPO on a low-dim version of the benchmarks in addition to the state-based versions.
 - In the introduction the authors mention that generative models like diffusion and consistency are helpful for RL because they allow the policy to represent complex behaviors. However, none of the tasks in the training suite are 'complex' enough that they couldn't be solved by a simple MLP policy. It would be nice to see perhaps some tasks where a policy that can sample multimodal actions shines above MLP-based policies.
 - Section 4 is confusing and there are several other minor presentation issues that I explain in the 'Questions' section

**Questions:**

Random Questions / Comments
 - I find the analysis related to figure 1 to be incomplete and hard to follow.
    - First, it would be helpful to explain somewhere in your paper what we can infer about a neural network based on its dormant ratio, which would make this experiment easier to understand. Your paper seems to assume the reader is very familiar with Sokar et. al. but this need not be the case
    - How big are the datasets? Where does the online dataset come from (ie what RL algorithm, how long was it trained, how often do you update the online dataset, etc)?
    - Why is there a difference between online and offline dormant ratio for halfcheetah and not walker2d? And how can you draw conclusions from this experiment when you only run two environments that seem to get different results?
 - Figure 2 experiment is also missing a lot of details
    - What AC algorithm does 'Q-loss' refer to?
    - Where does the data for this experiment come from?
    - Once again, I'm having trouble understanding how to interpret these plots or how you come to your conclusion.
 - Figure 3 has similar issues to the last two
 - The results as they are look nice but I have a few comments
    - I think the set of baselines is incomplete. It would be nice to see a SOTA on-policy method like PPO
    - All of the results show normalized scores, but you never explain how they are normalized, which makes them impossible to compare to other papers

Nit picks:
 - It would be nice if you could add one sentence in English describing what a dormant neuron is to section 3.3 for those who are not familiar with it. It would also be nice to explain why dormant ratio is important for the expressive capabilities of a network
 - Line 134: newtork

**Limitations:**

The authors address limitations and broader impacts of their work in the appendix.

---

> ### Author Rebuttal · Authors · 2024-08-06
>
> Many thanks for the careful review and constructive feedback. Please note our global response with the attached PDF. In the modified revision, we expound on the correlation between the dormant ratio and the performance of the policy network in Section 3.3 as shown in the global response.
> > These things should be able to work for both low-dimensional and image-based observations. Does this algorithm no longer show any benefit in these settings?
>
> In this work, we propose a proximal policy regularization method to stabilize the training of the consistency policy to improve the sample efficiency of visual RL. CP3ER indeed has the potential to be applied to state-based RL tasks. We supplement comparisons with SOTA methods[1]. The results are shown in Table 1 of the attached PDF. Compared to current SOTA methods, CP3ER demonstrates significant advantages in state-based RL tasks, further proving the generalization capability of CP3ER across different RL tasks.
> > It would be nice to see perhaps some tasks where a policy that can sample multimodal actions shines above MLP-based policies.
>
> Diffusion/consistency models build the policy distribution without modifying the structure of the policy network. Our policy network utilizes a simple MLP network, which is the same as the networks in Consistency-AC [2] and Diffusion-QL [3].
> > How big are the datasets? Where does the online dataset come from (ie what RL algorithm, how long was it trained, how often do you update the online dataset, etc)
>
> The pre-collected dataset from D4RL (medium-replay-v2) is used as the offline data, which is from the replay buffer of SAC, comprising 1 million transition tuples. For online training, we employ SAC as the baseline and sample data from the replay buffer to train consistency policy with consistency loss. The replay buffer is updated with each interacting step. For Figure 2, we use the same setting for consistency loss.
> > Why is there a difference between online and offline dormant ratio for halfcheetah and not walker2d? And how can you draw conclusions from this experiment when you only run two environments that seem to get different results?
>
> The dormant ratio change trends in the two subplots are the same, even though there are slight differences in the specific details. We speculate that the diversity of samples and actions in the halfcheetah dataset is lacking, leading the consistency policy to quickly fit the data distribution and overfit during offline training. In contrast, during the online training, due to the continuous updating of data, the policy network remains active. In the walker2d dataset, the diversity of the samples and action is high, requiring a certain training step to fit, resulting in the same as online learning.
>
> Due to dataset limitations (3 medium-replay datasets are available in D4RL), we conduct 3 tasks when analyzing the impact of offline/online settings on the dormant ratio of the policy. To analyze the effects of training losses and observations on the dormant ratio, we perform 5 different tasks, respectively. Curves not included in the main paper are available in the attached PDF, specifically in Fig. B and C.
> > What AC algorithm does 'Q-loss' refer to?
>
> Q-loss refers to Equation [2] in the paper, which is the loss used to update the policy.
> > (Figure 2 and 3) I'm having trouble understanding how to interpret these plots or how you come to your conclusion.
>
> When training the consistency policy using Q-loss, the variance in the dormant ratio of the policy network is quite large. This is because, under some random seed settings, the dormant ratio rapidly increases and remains at a relatively high level throughout. Subsequent training hardly changes the number of active neurons in the network, causing the policy to fall into a dormant where it outputs almost the same actions with different state inputs. With Fig. B and C in the attached PDF, this phenomenon indicates the performance descent. Compared to the state, the network's dormant ratio with the image quickly rises to high values with small variance, indicating severe degradation. With Fig. C in the attached PDF, we can observe that the policy degradation phenomenon of Consistency-AC in visual RL is severe.
> > It would be nice to see a SOTA on-policy method like PPO
>
> For visual RL, sample efficiency is a common issue. As an on-policy method, PPO does not exhibit notable sample efficiency, which is why it has not been extensively studied in visual RL. Instead, there is more focus on off-policy methods, such as shown in [4-7]. Additionally, in state-based RL, PPO performs poorly compared to other methods(Table 1 in the attached PDF). Furthermore, our motivation is to improve sample efficiency in visual RL, we believe there is no need to compare with on-policy methods like PPO in visual RL tasks.
> > All of the results show normalized scores, but you never explain how they are normalized, which makes them impossible to compare to other papers.
>
> We used the metrics recommended in [8] to analyze and compare the performance of algorithms.
> Additionally, we provide the unnormalized IQM curves in the appendix (Figure 12, 14, and 16) for comparison with results from other papers.
>
>
> [1] Boosting Continuous Control with Consistency Policy, AAMAS 2024.
>
> [2] Consistency Models as a Rich and Efficient Policy Class for Reinforcement Learning, ICLR 2024
>
> [3] Diffusion Policies as an Expressive Policy Class for Offline Reinforcement Learning, ICLR 2023
>
> [4] DrM: Mastering Visual Reinforcement Learning through Dormant Ratio Minimization, ICLR 2024(Figure 2, 11)
>
> [5] Mastering Visual Continuous Control: Improved Data-Augmented Reinforcement Learning, ICLR 2021
>
> [6] TACO: Temporal Latent Action-Driven Contrastive Loss for Visual Reinforcement Learning, NeurIPS 2023
>
> [7] Stabilizing Off-Policy Deep Reinforcement Learning from Pixels, ICML 2023
>
> [8] Deep Reinforcement Learning at the Edge of the Statistical Precipice, NeurIPS 2021.

---

> > ### Comment · Reviewer_ceLx · 2024-08-07
> > **Most concerns addressed**
> >
> > Thanks for running more experiments and addressing most of my concerns. I'll increase my score to a 5.
> >
> > My one major remaining concern is the following from my original review:
> > > In the introduction the authors mention that generative models like diffusion and consistency are helpful for RL because they allow the policy to represent complex behaviors. However, none of the tasks in the training suite are 'complex' enough that they couldn't be solved by a simple MLP policy. It would be nice to see perhaps some tasks where a policy that can sample multimodal actions shines above MLP-based policies.
> >
> > The authors pointed out that their policy is indeed still an MLP which indicates to me that I may have worded my complaint in a confusing manner. Here when I say 'MLP policy' I'm referring to a setup that directly maps an input state to a deterministic action, Gaussian distribution, or something similar, and examples of algorithms that use such policies would be vanilla SAC and DDPG. On the other hand, the authors use a consistency policy which, while still parameterized by an MLP, uses the consistency model framework in order to efficiently simulate the reverse diffusion process. The authors motivate this decision in the intro when they argue that consistency policies are more capable of addressing complex tasks that would not be solvable with a Gaussian policy. However, none of the tasks in the paper are too complex to be solved by a Gaussian, so I feel that it doesn't make sense to claim this setup is more capable of such tasks.

---

> > > ### Author Response · Authors · 2024-08-08
> > >
> > > Thank you for recognizing our work. In the introduction, we mentioned that unimodal policy such as Gaussian policy cannot model complex behavior. The complex behavior here refers to the complex exploration behavior caused by multi-modal reward landscapes. This situation exists in many tasks, which is one of the reasons why stochastic policy is needed. Previous works[1][2] have also shown that compared to unimodal policy distributions, multimodal distributions can escape from local optima easily, thereby achieving better sample efficiency and performance. Section 6.2.1 of the paper demonstrated the Gaussian distribution's limitations of exploration in this scenario with the 1D continuous bandit problem[2]. Figure 8 shows that due to the expressiveness to represent complex behavior, CP3ER achieves diverse exploration behaviors and better performance.
> > >
> > > In addition, experimental results(Section 6.1.1, Figure 6, and Section B.2 in Appendix, Figure 14 and 15.) on hard tasks of DeepMind control suite showed that the baseline DrQ-v2 using Gaussian policy hardly learned meaningful behavior within 5M steps, while CP3ER achieved a relatively good performance, indicating that in high-dimensional continuous action spaces, CP3ER has stronger exploration ability compared to Gaussian policy.
> > >
> > > [1] Reinforcement Learning with Deep Energy-Based Policies, ICML 2017
> > >
> > > [2] Reparameterized Policy Learning for Multimodal Trajectory Optimization, ICML 2023

---

> > > ### Author Response · Authors · 2024-08-12
> > >
> > > Thank you again for your time in helping us improve our work. We hope the reply can address your concerns. We sincerely appreciate your recognition of our contribution and vote to accept our work!

---

### Official Review · Reviewer_UMZF · 2024-07-10

**Soundness:** 3
**Presentation:** 3
**Contribution:** 2
**Rating:** 5
**Confidence:** 3

**Summary:**

This paper proposes a novel method that employs a consistency model for policy parameterization in online visual reinforcement learning settings. It initially identifies the issues of prior consistency actor-critic methods in high-dimensional observation online RL settings through the lens of dormant ratio. Subsequently, it proposes to incorporate entropy regularization alongside a prioritized sampling strategy to regularize the policy. Empirical experiments on 21 tasks across both the DeepMind Control Suite and MetaWorld demonstrate the effectiveness of the proposed method.

**Strengths:**

- Overall, the paper is well-organized and presented.
- The analysis of the limitations of existing consistency policy training in visual RL settings is thorough and convincing.
- The experimental results across a wide range of tasks are promising compared to previous state-of-the-art methods.
- The paper includes a good set of ablation studies to support the design choices of the proposed algorithm.

**Weaknesses:**

- Lack of baseline comparison in the experiments
- The scope of the paper is limited to online and visual deep reinforcement learning
- The proposed modification is straightforward compared with existing algorithms.

**Questions:**

- Although CP3ER is mainly designed for online visual RL, the proposed modifications (entropy regularization, prioritized replay, Gaussian mixture distribution of value functions) do not seem to be limited to visual observation settings. It would be interesting to see how the proposed algorithm performs in state-based RL settings.
- For the 8 hard DeepMind Control Suite tasks and also Metaworld tasks, the authors compare CP3ER with TACO/ALIX/DrQ-v2 but not with DrM, which was reported in the original DrM paper to achieve good empirical performance on these tasks. This comparison here is missing.
- For the analysis (Figures 2, 3, 9), it would be beneficial to include the episodic return in the plots so that readers can understand the numerical performance of the policy in addition to the policy network’s dormant ratio, which is an intrinsic factor not necessarily correlated with the policy’s episodic return.
- The conclusion drawn from lines 202-203, “Therefore, we can infer that visual RL will exacerbate the instability of consistency policy training caused by the Q-loss under the actor-critic framework,” does not seem convincing. Even in ordinary policy

---

> ### Author Rebuttal · Authors · 2024-08-06
>
> Thank you for your detailed review and helpful suggestions. Please note the global response and the attached PDF.
> > The proposed modification is straightforward compared with existing algorithms.
>
> The key challenge lies in addressing the instability during the training of consistency policy. By analyzing changes in the dormant ratio of the policy network, we observed a policy degradation phenomenon and speculate that this may be due to the instability of the data distribution and the score function represented by the Q-function. Therefore, we introduce entropy regularization. From the perspective of RL, our goal is to imbue the policy with specific attributes by modifying the loss, such as maximizing entropy to boost sample efficiency. From the perspective of matching distributions, entropy regularization provides a prior for the target data distribution, with the Q-function weighting this prior, thus stabilizing the optimization objective of the consistency model.
> > It would be interesting to see how the proposed algorithm performs in state-based RL settings.
>
> CP3ER can indeed be applied to state-based RL. Following [1], we selected 6 challenging tasks from the DeepMind Control Suite(DMC) environment to evaluate the methods. Table 1 in the attached PDF shows the results of different methods within 500k interacting steps. Compared to commonly used RL methods, CP3ER has significant performance advantages and also outperforms the current SOTA diffusion model-based method CPQL.
>
> > For the 8 hard DeepMind Control Suite tasks and also Metaworld tasks, the authors compare CP3ER with TACO/ALIX/DrQ-v2 but not with DrM, which was reported in the original DrM paper to achieve good empirical performance on these tasks. This comparison here is missing.
>
> DrM is indeed an excellent method for visual RL. In fact, we considered DrM in the appendix. Figures 14 and 15 show the comparison of DrM on DMC hard tasks. Figures 16 and 17 show the comparison results on Meta-world tasks. We did not include these results in the main part of the paper because when we reproduced DrM results using the official code[2], we could not achieve the performance claimed on Meta-world tasks. To obtain a fairer comparison, we compared our method and the reproduced results with the curves from [3], as shown in Table 2 of the attached PDF. Even compared to the results in the DrM paper, our method achieves comparable or even better performance. We have also included performance curves for several methods on DMC-hard and Meta-world tasks in the attached PDF(Fig. A, the left part shows curves on DMC-hard tasks and the right part shows curves on Meta-world tasks.).
> > For the analysis (Figures 2, 3, 9), it would be beneficial to include the episodic return in the plots so that readers can understand the numerical performance of the policy in addition to the policy network’s dormant ratio, which is an intrinsic factor not necessarily correlated with the policy’s episodic return.
>
> Thank you very much for your suggestion. We have added an explanation of the relationship between the dormant ratio and policy performance after Section 3.3 in the revised version as shown in the global response.
>
> However, it is important to note that the dormant ratio is a measure of the capacity and the expressiveness of the network, and it is not a sufficient and necessary condition for assessing policy performance. A policy network with a low dormant ratio may have a low episode return. For example, when trained with supervised learning (such as consistency loss), the network typically exhibits a low dormant ratio, but since its goal is to fit the data distribution rather than to maximize average returns, its policy performance may not necessarily be superior. To adequately assess the policy degradation phenomenon caused by Q-loss, we compared the performance of the consistency policy trained with Q-loss to SAC (the online data collection policy required when training with consistency loss). We also compared the dormant ratios of the policy networks in SAC and redrew Figure 2 (Figure B middle and right, and Figure C in the attached PDF). Consistency policy with Q-loss has a higher dormant ratio compared to those trained with consistency loss and SAC.
>
> We revised Figure 3 and 9 from the original paper with average returns as the other axis. The modifications are shown in Figures D and E in the attached PDF. The results from the figures indicate that generally, the higher the dormant ratio of the policy network, the poorer the performance of the policy. Visual RL tasks exacerbate the degradation phenomenon of consistency policies, resulting in poor policy performance.
> > The conclusion drawn from lines 202-203, “Therefore, we can infer that visual RL will exacerbate the instability of consistency policy training caused by the Q-loss under the actor-critic framework,” does not seem convincing. Even in ordinary policy
>
> In the third part of Section 4, we analyzed consistency policy, particularly whether Consistency-AC is suitable for visual RL tasks. Figure 3 shows the dormant ratio of the policy network under different observations. Compared to state inputs, the network's dormant ratio rapidly rises to very high values with visual inputs, and the variance is small. This indicates that this phenomenon will stably exist, unaffected by random seeds.  Compared to state-based RL tasks, we can see that the policy degradation phenomenon of Consistency-AC in visual RL is quite severe. Consistency-AC adopts a Q-loss under the Actor-Critic framework. Thus, visual RL exacerbates the instability of consistency policies. In the modified revision, we add the explanation in 3.3 discussing the relationship between the dormant ratio and the policy performance.
>
> [1] Boosting Continuous Control with Consistency Policy, AAMAS 2024
>
> [2] https://github.com/XuGW-Kevin/DrM
>
> [3] DrM: Mastering Visual Reinforcement Learning through Dormant Ratio Minimization, ICLR 2024

---

> ### Author Response · Authors · 2024-08-12
>
> We hope the reply can address your concerns. We sincerely appreciate your recognition of our contribution and vote to accept our work!

---

### Official Review · Reviewer_Xhh1 · 2024-07-12

**Soundness:** 3
**Presentation:** 3
**Contribution:** 3
**Rating:** 7
**Confidence:** 4

**Summary:**

The paper addresses challenges in visual RL with high-dimensional state spaces, specifically focusing on improving sample efficiency and training stability. It introduces a novel method called CP3ER, which incorporates sample-based entropy regularization and prioritized proximal experience regularization to stabilize policy training and enhance sample efficiency. The proposed CP3ER achieves state-of-the-art performance in 21 tasks across the DeepMind control suite and Meta-world, demonstrating the effectiveness of applying consistency models to visual RL. The paper identifies and addresses the instability issues caused by the Q-loss in the actor-critic framework and the non-stationary distribution of online RL data. Experimental results show that CP3ER outperforms existing methods without relying on additional exploration strategies or auxiliary losses.

**Strengths:**

1. The introduction of CP3ER, which combines consistency models with prioritized proximal experience regularization, presents a novel approach to addressing challenges in visual RL, enhancing both sample efficiency and training stability.
2. The paper provides comprehensive empirical evidence demonstrating that CP3ER achieves state-of-the-art performance across a wide range of tasks in the DeepMind control suite and Meta-world.
3. The paper provides a thorough analysis of the impact of non-stationary distributions and the actor-critic framework on consistency policy, offering insights into the underlying mechanisms that affect policy training stability.

**Weaknesses:**

In general, the paper is well-written and easy to follow. However, there is no illustration or theoretical support for using Eq. (8) for sampling weight. It would be better for the authors to consider more theoretical analysis on the solution design.

**Questions:**

Why using eq. (8) for sampling weight? Can other functions that shares a similar look demonstrated in Fig 4(b) also work?

**Limitations:**

Limitations have been discussed in the paper.

---

> ### Author Rebuttal · Authors · 2024-08-06
>
> Thank you very much for your recognition of our work.
>
> > Why using eq. (8) for sampling weight? Can other functions that shares a similar look demonstrated in Fig 4(b) also work?
>
> In fact, Equation (8) is an empirical formula inspired by the Sigmoid function, and it has been improved to meet the desired properties. We aim to sample the most recently collected data with a high probability during the sampling process while sampling older data with a relatively lower probability. If other formulas can satisfy the required properties (Fig. 4(b)), they may also be used to generate sampling weights.

---

### Official Review · Reviewer_TpQn · 2024-07-12

**Soundness:** 3
**Presentation:** 3
**Contribution:** 3
**Rating:** 6
**Confidence:** 3

**Summary:**

The paper analyzes the problems faced by extending consistency policy to visual RL under the actor-critic framework and discovers the phenomenon of the collapse of consistency policy during training under the actor-critic framework by analyzing the dormant rate of the neural networks. The authors propose a consistency policy with prioritized proximal experience regularization (CP3ER), which employs entropy regularization to constrain policy behavior. The experiments evaluate CP3ER with recent baselines (DrQ-v2, ALIX, TACO), on 21 visual control tasks.

**Strengths:**

- The paper is well-written and organized, and includes a thorough discussion of the relevant related works.
- Sec. 4 analyzes the Consistency Actor-Critic from the perspective of dormant rates, which is very interesting.
- The experimental results showcase impressive performance gains over the state-of-the-art on a wide range of robotics tasks (e.g., DM-Control, Meta-World). Ablations are also provided to delineate the impact of each modification.

**Weaknesses:**

- The method is not directly applicable to domains with discrete action spaces.
- The experiments do not compare with other consistency RL methods.

**Questions:**

- According to [1], dormant rates in reinforcement learning and supervised learning represent different meanings. Can you explain in detail how dormant rates affect the expression ability of the consistency policy during training? For example, what does a high dormant rate indicate? What does an increasing dormant rate signify? And why?
    - [1] Sokar, Ghada, et al. "The dormant neuron phenomenon in deep reinforcement learning." *International Conference on Machine Learning*. PMLR, 2023.
- In Sec. 4, Fig. 3, since Q-loss under the actor-critic framework will destabilize the consistency policy training, why was only the Q-loss used during the training process?
- How do the computational cost and runtime compare with other baselines?

**Limitations:**

See the weaknesses and questions section.

---

> ### Author Rebuttal · Authors · 2024-08-06
>
> Thank you for your detailed review and valuable comments. Please note the global response with the attached PDF.
>
> > The method is not directly applicable to domains with discrete action spaces.
>
> Indeed, our method has not yet been extended to discrete action spaces because we use consistency models to model the policy, and most diffusion/consistency models are currently applied in continuous state spaces. Therefore, our proposed method is also subject to this limitation. However, to extend it to discrete action spaces, one possible approach could be to binarize the discrete actions and then map them to continuous spaces, as suggested in [1]. This could be an interesting and worthwhile direction for future exploration.
>
> > The experiments do not compare with other consistency RL methods.
>
> Our proposed method aims to address the issue of sample efficiency in visual RL. To the best of our knowledge, as of the time of paper submission, CP3ER is the only one to apply diffusion/consistency models to online visual RL policy. This is why we have not compared it with other RL methods based on diffusion models or consistency models. In the realm of state-based RL, there are already some methods based on diffusion models or consistency models, and we have compared our method with them. The results are shown in Table 3 of the attached PDF. Following the settings in [2], we compared CP3ER with Consistency-AC and online Diffusion-QL. As can be seen from the results, although our method is designed to visual RL problems, it still demonstrates significant advantages when extended to state-based RL tasks compared to existing methods based on diffusion or consistency models.
>
> > Can you explain in detail how dormant rates affect the expression ability of the consistency policy during training? For example, what does a high dormant rate indicate? What does an increasing dormant rate signify? And why?
>
> The dormant ratio of a neural network represents the proportion of neurons that are inactive. The higher dormant ratio indicates fewer active neurons in the neural network, implying the network's capacity and expressiveness are damaged. The policy network's dormant ratio in RL is related to episode return. A higher dormant ratio hints at lazy action in RL with lower episode returns. Conversely, when the performance is good, the policy network is usually more active, and the dormant ratio is usually lower. This phenomenon has been observed in several studies [3-6].
>
> > In Sec. 4, Fig. 3, since Q-loss under the actor-critic framework will destabilize the consistency policy training, why was only the Q-loss used during the training process?
>
> In the third part of Section 4, our main objective is to investigate whether consistency policy is suitable for visual RL tasks. Consistency-AC [2], a typical representative of consistency policy, uses only Q-loss during training. Therefore, we maintained the same settings in this part of the experiment. Fig. 3 illustrates the impact of different observations on the dormant ratio of the consistency policy network. The phenomena in the figure show that consistency policy in visual RL faces a very high dormant ratio, making policy training challenging. From Fig. 3, we can also conclude that Consistency-AC is not suitable for visual RL learning tasks.
>
> > How do the computational cost and runtime compare with other baselines?
>
> We evaluated the training and inference time of CP3ER separately, comparing it with the methods mentioned in the paper. The GPU used is 2080Ti and the batch size for training is 256. The results are shown in the following table. Compared to TACO, our proposed CP3ER does not show a significant increase in training time and inference time costs. During training, the additional time cost of CP3ER compared to DrQv2 is mainly attributed to the computation of the regularization loss term.
>
>
> |       | DrQv2 | ALIX | TACO | DrM | CP3ER |
> | ----------- | ----------- | ----------- | ----------- | ----------- | ----------- |
> |Training Time per Batch (s) | 0.0339 | 0.0332 | 0.1919 | 0.04075 | 0.04668 |
> | Inference Time per Step (s) | 0.00132 | 0.00141 | 0.00131 | 0.00184 | 0.00218 |
>
>
> [1] Structured Denoising Diffusion Models in Discrete State-Spaces, ICLR 2023.
>
> [2] Consistency Models as a Rich and Efficient Policy Class for Reinforcement Learning, ICLR 2024
>
> [3] DrM: Mastering Visual Reinforcement Learning through Dormant Ratio Minimization, ICLR 2024(Figure 2, 11)
>
> [4] In deep reinforcement learning, a pruned network is a good network, Arxiv 2024(Figure 11)
>
> [5] The Dormant Neuron Phenomenon in Deep Reinforcement Learning, ICML 2023 (Figure 9, 11, 16)
>
> [6] Pretrained Visual Representations in  Reinforcement Learning, Arxiv 2024 (Figure3, 4)

---

> ### Author Response · Authors · 2024-08-12
>
> We hope the reply can address your concerns. We sincerely appreciate your recognition of our contribution and vote to accept our work!

---

> > ### Comment · Reviewer_TpQn · 2024-08-13
> >
> > I appreciate the authors' clarifications and still remain in favor of acceptance.

---

### Author Rebuttal · Authors · 2024-08-06

**Please see the attached one-page PDF for additional experimental results.**

We would like to express our sincere gratitude for the efforts and valuable feedback provided by all the reviewers. We are very pleased that our work has been recognized by the reviewers, and we have also noted some points that have been commonly raised:

## Can CP3ER extend to state-based RL?

Both reviewers *UMZF* and *ceLx* have raised concerns regarding whether our proposed CP3ER can be generalized to state-based RL tasks. Our initial motivation was to address the issue of sample efficiency in visual RL. The high expressiveness of the policy helps to improve exploration and thereby enhances sample efficiency. Directly introducing diffusion/consistency models into visual RL results in severe policy degradation, preventing policy training. Therefore, we proposed a proximal policy regularization method to stabilize the training of consistency policies in visual RL. We also recognize that CP3ER indeed has the potential to be applied to state-based RL tasks. Consequently, we have included additional experiments in the attached PDF (Table 1 and Table 3) comparing CP3ER with SOTA methods in state-based RL. We found that CP3ER still exhibits advantages in state-based tasks. However, these advantages are not as pronounced as in visual RL tasks, because while policy degradation also occurs in state-based tasks using diffusion/consistency models, it is not as severe as in visual RL. These results also support the discussion in the last paragraph of Section 4 of our paper.

## What is the relationship between dormant ratio, policy degradation, and policy performance?

In Section 4, reviewers *TpQn*, *UMZF*, and *ceLx* care about how the conclusion of the policy degradation derived from the changes in the dormant ratio of the policy network. It must be acknowledged that in writing the paper, we neglected that readers might not be familiar with the concept of dormant ratio in RL. As introduced in [3], the dormant ratio of a neural network indicates the proportion of inactive neurons and is typically used to measure the activity of the network. A higher dormant ratio implies fewer active neurons in the network, implying the network's capacity and expressiveness are damaged. In RL, the episode return is closely related to the dormant ratio of the policy network. A higher dormant ratio results in more lazy action outputs, inactive agent behavior, and lower episode returns; conversely, when policy performance is good, the policy network is usually more active, and the dormant ratio is typically lower. This phenomenon has been reported in [3-6]. Based on these findings and the results in Section 4, Q-loss is identified as the primary reason for the increase in the dormant ratio of consistency policy. An increase in the dormant ratio often signifies a decline in policy performance (We have added additional curves of dormant ratio and performance in Fig. B, C, and D of the attached PDF. ), thus indicating policy degradation. According to the results in Figure 3 of the paper (Figure D in the attached PDF. Considering the page limit, the tasks that have already appeared in the main paper have not been included in the attached PDF.), it can be concluded that consistency policy in visual RL tasks faces severe policy degradation.

In the modified revision, we expound on the correlation between the dormant ratio and the performance of the policy network in Section 3.3 as follows:

> As introduced in [3], the dormant ratio of a neural network indicates the proportion of inactive neurons and is typically used to measure the activity of the network. A higher dormant ratio implies fewer active neurons in the network, implying the network's capacity and expressiveness are damaged. In RL, the episode return is closely related to the dormant ratio of the policy network. A higher dormant ratio results in more lazy action outputs, inactive agent behavior, and lower episode returns; conversely, when policy performance is good, the policy network is usually more active, and the dormant ratio is typically lower. This phenomenon has been reported in [3-6].

Regarding the other issues raised by the reviewers, we provide detailed responses below. We strive to address all concerns raised by the reviewers. If there are any further questions, we are also very pleased to discuss them.

We have made revisions to the paper addressing certain issues. In the revised version, changes will be highlighted in blue.

[1] Boosting continuous control with consistency policy, AAMAS 2024

[2] Consistency Models as a Rich and Efficient Policy Class for Reinforcement Learning, ICLR 2024

[3] The dormant neuron phenomenon in deep reinforcement learning, ICML 2023 (Figure 9, 11, 16)

[4] DrM: Mastering Visual Reinforcement Learning through Dormant Ratio Minimization, ICLR 2024(Figure 2, 11)

[5] In deep reinforcement learning, a pruned network is a good network, Arxiv 2024(Figure 11)

[6] Pretrained Visual Representations in  Reinforcement Learning, Arxiv 2024 (Figure 3, 4)

---

### Decision · Program_Chairs · 2024-09-25

**Decision:**

Accept (poster)

**Comment:**

There is a unanimous decision for acceptance, congratulations! The sample-based entropy regularization and prioritized proximal experience regularization make sense, and the experimental comparisons are extensive and convincing on the DM and Meta-World environments.